# Efficient Genome Editing Using the T2A-Coupled Co-Expression of Two ZFN Monomers

**DOI:** 10.3390/ijms26157602

**Published:** 2025-08-06

**Authors:** Shota Katayama, Takashi Yamamoto

**Affiliations:** 1Genome Editing Innovation Center, Hiroshima University, Higashi-Hiroshima 739-0046, Japan; tybig@hiroshima-u.ac.jp; 2Division of Integrated Sciences for Life, Graduate School of Integrated Sciences for Life, Hiroshima University, Higashi-Hiroshima 739-8526, Japan

**Keywords:** genome editing, ZFN, 2A peptide

## Abstract

Genome editing is commonly used in biomedical research. Among the genome editing tools, zinc finger nucleases (ZFNs) are smaller in size than transcription activator-like effector nucleases (TALENs) and CRISPR-Cas9. Therefore, ZFNs are easily packed into a viral vector with limited cargo space. However, ZFNs also consist of left and right monomers, which both need to be expressed in the target cells. When each monomer is expressed separately, two expression cassettes are required, thus increasing the size of the DNA. This is a disadvantage for a viral vector with limited cargo space. We herein showed that T2A-coupled ZF-ND1 monomers were co-expressed from a single expression cassette and that the corresponding ZF-ND1s efficiently cleaved the target DNA sequences. Furthermore, the total amount of transfected plasmid DNA was reduced by half, and genome editing efficiency was equivalent to that of two separate ZF-ND1 monomers. This study provides a promising framework for the development of ZFN applications.

## 1. Introduction

Genome editing tools are widely used for biomedical research. Among them, CRISPR-Cas9 is the most frequently used worldwide [1]. However, CRISPR-Cas9 patents are active until 2033 [2,3], and thus, high patent royalties are required for industry applications. In contrast, the patents for zinc finger nucleases (ZFNs) ended in 2020 [4]. Moreover, a previous study demonstrated that the sizes of assembled ZFs (0.3–0.6 kbp) were smaller than those of assembled transcription activator-like effectors (encoded in 1.7–2 kbp) or SpCas9 (4.1 kbp) [5]. Due to their sizes, the packing of ZFNs into the limited cargo space (~4.6 kbp) of adeno-associated virus (AAV) vectors is easily achieved [6,7]. Therefore, AAV/ZFN vectors are a powerful platform for the in vivo delivery of gene therapies, and the application of ZFNs in translational research has great potential.

ZFNs comprise a DNA-binding moiety and catalytic moiety, with assembled ZF proteins functioning as the former, and ND1 heterodimer nucleases (DDD/RRR) [8] as the latter. The DNA cleavage activity of a ND1 heterodimer nuclease (DDD/RRR) was previously shown to be higher than that of a FokI heterodimer nuclease (ELD/KKR) [8]. We successfully developed various ZF-ND1s by linking the ND1 heterodimer nuclease (DDD/RRR) to assembled ZF proteins [9].

ZF-ND1s consist of two monomers, a left ZF-ND1RRR monomer and a right ZF-ND1DDD monomer. In a previous study [9], each monomer was expressed from a single expression cassette, and thus, two expression cassettes were required. Therefore, a large size of DNA was needed for expressing both monomers, which was disadvantageous for using an AAV vector. We hypothesized that two monomers could be expressed from a single expression cassette using a 2A peptide.

A 2A peptide has a conserved DxExNPGP motif that promotes ribosomal skipping upon translation [10]. There are four kinds of 2A peptides: P2A, T2A, E2A, and F2A [11,12]. Among them, P2A and T2A are the most efficient in mammalian cells [10,13,14]. Therefore, two ZFN monomers should be coupled with a P2A or T2A peptide. In the case of ZF-FokI [15], T2A has already been used for the coupling of two ZF-FokI monomers [16,17,18].

In the present study, we developed T2A-coupled ZF-ND1 monomers, which successfully cleaved the target DNA with the same efficiency as two separate monomers. Furthermore, the total amount of transfected plasmid DNA was reduced by half, and the genome editing efficiency was equivalent to that achieved using separate monomers.

## 2. Results

### 2.1. Design of ZF-ND1 Expression

We previously developed a new type of ZFN, namely ZF-ND1 [9]. ZF-ND1s consist of either left or right 6-finger ZFs and the ND1 nuclease domain, recognizing 36 DNA bases each and cleaving the spacer region. In the previous study, ZF-ND1 was expressed from two different plasmids, one of which codes left ZF (ZF(L)) and ND1RRR, whereas the other codes right ZF (ZF(R)) and ND1DDD (Figure 1a). In this study, to co-express ZF(L)-ND1RRR and ZF(R)-ND1RRR from a single expression cassette in a single plasmid, we designed a single plasmid coding for ZF(L)-ND1RRR and ZF(R)-ND1DDD linked using a P2A or T2A peptide (Figure 1b). Because the cleaved P2A or T2A peptide is added to ND1RRR, we speculated that it may influence the efficiency of genome editing.

### 2.2. T2A Peptide Enables Efficient Genome Editing with a Single Plasmid

To clarify whether the cleaved P2A or T2A peptide influences the efficiency of genome editing, we constructed a single plasmid coding for 2027L-ND1RRR and 2027R-ND1DDD linked using the P2A or T2A peptide (Figure 2a). We then transfected these plasmids into HEK293T cells. Genomic DNA was extracted 72 h later and subjected to a T7EI assay. The results obtained showed that the genome editing efficiency of T2A-coupled ZF-ND1 was similar to that of the separate monomers (Figure 2b), indicating that the cleaved T2A peptide does not influence the efficiency of genome editing. Conversely, the cleaved P2A peptide greatly influenced and reduced the efficiency of genome editing (Figure 2b). Given that ZF-ND1 linked with T2A peptide is expressed from a single plasmid, we hypothesized that the total amount of transfected plasmid DNA may be reduced to half of that required using separate monomers. To address this, we reduced the total amount of plasmids containing P2A or T2A peptide to half that used for separate monomers. The subsequent T7EI assay showed that the total amount of plasmid DNA containing T2A was reduced by half, and genome editing efficiency was almost equivalent to that using separate monomers (Figure 2c). To investigate whether off-target mutations were induced by T2A-coupled ZF-ND, two off-target sites of the highest potential of each ZF-ND1, ranked using Nucleotide BLAST Identities, were selected. These sites were amplified and then examined using the T7EI assay. The results obtained showed that there were no additional bands among untransfected cells or cells transfected with the separate monomers or T2A-coupled ZF-ND1 (Appendix A), indicating the absence of mutations at these off-target sites (Appendix A).

### 2.3. Efficient Genome Editing with a Single Plasmid Encoding Other ZF-ND1s

To clarify whether the T2A peptide is effective on other ZF-ND1s, we constructed a single plasmid containing 9299L(v2)-ND1RRR and 9299R-ND1DDD linked with P2A or T2A peptide (Figure 3a). Plasmids were transfected into HEK293T cells. The AAVS-1 site was amplified by PCR 72 h later, and the resulting DNA fragments were examined using the T7EI assay. The results obtained showed that the genome editing efficiency of T2A-coupled ZF-ND1 was equivalent to that of the separate monomers (Figure 3b), indicating that the T2A peptide was effective for other ZF-ND1s. Furthermore, we reduced the total amount of plasmids containing P2A or T2A peptide to half compared with the amount required using separate monomers. A T7EI assay showed that the total amount of plasmid DNA containing T2A could be reduced to half, and the genome editing efficiency remained equivalent to that observed for separate monomers (Figure 3c). To investigate whether off-target mutations were induced by T2A-coupled ZF-ND1, two off-target sites of the highest potential of each ZF-ND1 were amplified by PCR. The DNA fragments obtained were then examined using the T7EI assay. There were no additional bands among untransfected cells or cells transfected with separate monomers or T2A-coupled ZF-ND1 (Appendix A), indicating the absence of mutations at these off-target sites (Appendix A).

### 2.4. Lower Doses of ZF-ND1 Plasmids and Another Cell Line

To clarify whether the 2A designs outperformed the two-plasmid design at a lower dose, we reduced the total amount of plasmid DNAs transfected into cells. The T7EI assay showed that the 2A designs outperformed the two-plasmid design at a lower dose (Appendix A).

To investigate whether the T2A peptide enabled efficient genome editing with a single plasmid on another cell line, we transfected the single plasmid into Jurkat cells. Seventy-two hours after transfection, genomic DNA was extracted and subjected to the T7EI assay. The results obtained showed that the total amount of plasmid DNA containing T2A was reduced by half, and the efficiency of genome editing was equivalent to that obtained using separate monomers (Appendix A).

## 3. Discussion

In this study, the P2A peptide greatly reduced the genome editing efficiency (Figure 2b,c and Figure 3b,c). P2A peptide is cleaved into 21 and 1 amino acids (Figure 1b). The 21 amino acids were added to ND1RRR and lowered the genome editing efficiency, indicating that the 21 amino acids inhibit the dimerization of the ND1 nuclease. In contrast, T2A peptide is cleaved into 20 and 1 amino acids (Figure 1b). The 20 amino acids were added to ND1RRR and did not reduce the genome editing efficiency, indicating that the 20 amino acids do not inhibit the dimerization of the ND1 nuclease. Even in the case of the FokI nuclease [15], T2A peptide does not reduce the genome editing efficiency and does not inhibit the dimerization of the FokI nuclease [16,17,18]. Therefore, T2A peptide is the most suitable for heterodimer nucleases.

For the AAV/ZFN vector, the number of expression cassettes is critical. Increasing the number of cassettes results in larger DNA, and thus, all ZFN components should be driven under a single expression cassette. T2A-coupled co-expression of two ZFN monomers is an effective strategy for achieving this objective, enabling various in vivo ZFN applications.

## 4. Materials and Methods

### 4.1. ZF Design

Two ZF prototypes, namely 2027L+2027R (V109K) and 9299L(v2)+9299R, were created in a previous study [9] and used in this study.

### 4.2. Vector Construction

2027L or 9299L(v2)-ND1RRR plasmid [9] was amplified by inverse PCR with Herculase DNA polymerase (Agilent, Santa Clara, CA, USA). The resulting fragment was the backbone for In-Fusion cloning. 2027R or 9299R-ND1DDD plasmid [9] was amplified by PCR. The resulting fragment was the insert for In-Fusion cloning. The backbone and insert were assembled using the In-Fusion HD Cloning Kit (TaKaRa, Kusatsu, Japan). Plasmid transformation was conducted using NEB Stable Competent Cells (NEB) (Ipswich, MA, USA), and the QIAprep Spin Miniprep Kit (QIAGEN, Hilden, Germany) was employed to purify plasmids. Nucleotide sequences were validated by Sanger sequencing. Primers are shown in Appendix A.

### 4.3. Cell Culture, DNA Transfection, and DNA Electroporation

HEK293T cells were cultured at 37 °C under 5% CO_2_ in DMEM (Nacalai Tesque, Kyoto, Japan) containing 10% FBS (HyClone, Logan, UT, USA), 100 units/mL of penicillin, and 100 µg/mL of streptomycin (Nacalai Tesque, Kyoto, Japan). Lipofectamine 2000 (Life Technologies, Carlsbad, CA, USA) and Opti-MEM (Life Technologies) were used as described by the manufacturer for the transfection of plasmids at the following concentrations into 0.5 × 10^5^ HEK293T cells in a 24-well plate: 150 ng each of the left ZF-ND1RRR and right ZF-ND1DDD plasmids, 75 ng each of the left ZF-ND1RRR and right ZF-ND1DDD plasmids, 25 ng each of the left ZF-ND1RRR and right ZF-ND1DDD plasmids, or 50, 150, or 300 ng of the left ZF-ND1RRR-2A-right ZF-ND1DDD plasmid.

Jurkat cells were cultured at 37 °C under 5% CO_2_ in RPMI1640 (Nacalai Tesque) containing 10% FBS (HyClone), 100 units/mL of penicillin, and 100 µg/mL of streptomycin (Nacalai Tesque). An OC-25X3 cuvette (MaxCyte) (Rockville, MD, USA), MaxCyte Buffer (MaxCyte), and the MaxCyte ATX system (MaxCyte) were used for electroporation according to the manufacturer’s instructions. Plasmids at the following concentrations were transfected into 1 × 10^6^ Jurkat cells: 1 µg each of the left ZF-ND1RRR and right ZF-ND1DDD plasmids, 0.5 µg each of the left ZF-ND1RRR and right ZF-ND1DDD plasmids, or 1 µg of the left ZF-ND1RRR-2A-right ZF-ND1DDD plasmid.

### 4.4. Transient Cold Shock

The temperature of the culture was reduced to 30 °C 24 h after plasmid transfection, and cells were cultured for a further 48 h. Cells were then harvested and examined using the T7EI assay.

### 4.5. T7EI Assay

The QIAamp DNA Mini Kit (QIAGEN) was employed to extract genomic DNA before PCR. The KOD One PCR Master Mix (Toyobo, Osaka, Japan) with an appropriate primer set (Appendix A) amplified the target site. Following the purification of PCR products using the MinElute PCR Purification Kit (QIAGEN), denaturing and reannealing reactions were performed at 95 °C for 5 min, at 95 to 85 °C at −2.0 °C/s, at 85 to 25 °C at −0.1 °C/s, and then continuously at 4 °C. The products obtained from incubating heteroduplexed PCR products with T7E1 (NEB) at 37 °C for 30 min were examined by electrophoresis on a 2% agarose gel and visualized using Gel Red. ImageJ 1.54p (NIH, Bethesda, MD, USA) was used to assess the band intensities of PCR amplicon and cleavage products. Efficiency was calculated as follows: % gene modification = 100 × (1 − (1− fraction cleaved)^1/2^).

### 4.6. Off-Target Analysis

Off-target analysis was performed as previously described [9]. In brief, off-target candidate sites for ZF-ND1 pairs were identified using Nucleotide BLAST (https://blast.ncbi.nlm.nih.gov/Blast.cgi, accessed on 2 June 2025). The T7E1 assay was conducted to assess PCR-amplified candidate sites. The primers used are listed in Appendix A.

## Figures and Tables

**Figure 1 ijms-26-07602-f001:**
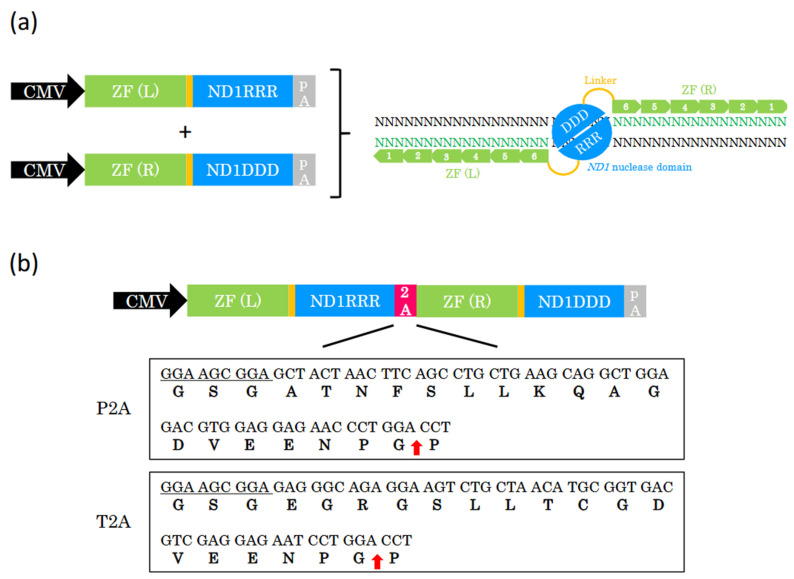
The design of the 2A-coupled co-expression of two ZF-ND1 monomers. (**a**) Schematic of two different expression cassettes and ZF-ND1. ZF (L)-ND1RRR and ZF (R)-ND1DDD are separately expressed from each expression cassette. ZF (L) and ZF (R) (shown in green boxes with white numbers) recognize the target DNA bases (shown in green text), and the ND1 heterodimer nuclease (DDD/RRR) cleaves the spacer region between their recognition sites. CMV, CMV promoter; ZF (L), left zinc finger; ZF (R), right zinc finger; pA, polyA signal. (**b**) The schematic of 2A-coupled co-expression of two ZF-ND1 monomers and P2A and T2A peptides. CMV drives two ZF-ND1 monomers connected with a 2A peptide. DNA and corresponding amino acid sequences of P2A and T2A. Underlined DNA sequences encode GSG, which was added to improve cleavage efficiency. Red arrows indicate the cleaved site in each 2A peptide.

**Figure 2 ijms-26-07602-f002:**
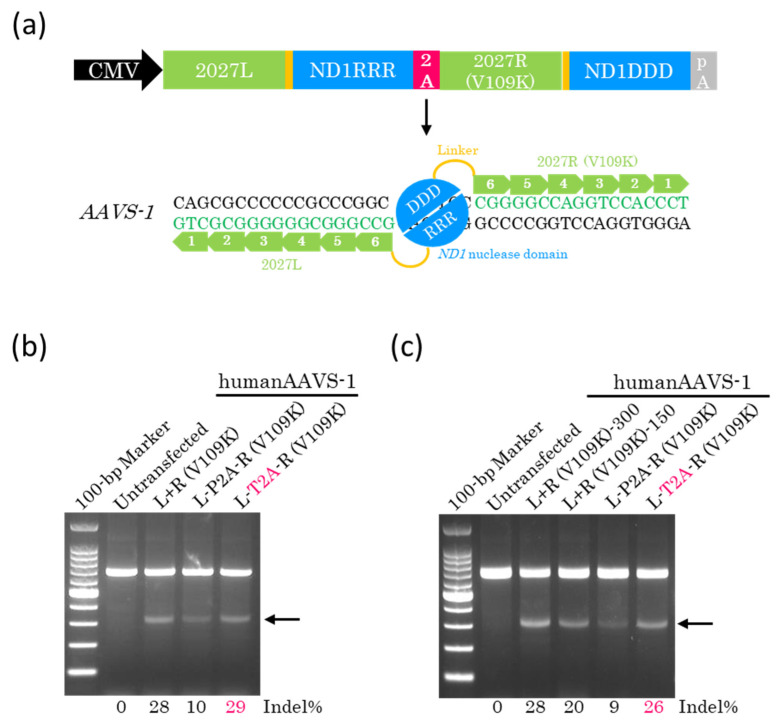
The development of the 2A-coupled co-expression of two ZF-ND1 monomers. (**a**) The schematic of the 2A-coupled co-expression of two ZF-ND1 monomers targeting *AAVS-1*. 2027L and 2027R (V109K) recognize their target DNA bases, and then the ND1 heterodimer nuclease (DDD/RRR) cleaves the spacer region between their recognition sites. (**b**,**c**) The T7EI assay. Gel images of T7EI-treated PCR products amplified from the target site. Arrows show the cleaved DNA band. L+R (V109K) indicates the separately expressed ZF-ND1 monomers. L-P2A-R (V109K) and L-T2A-R (V109K) indicate the P2A- and T2A-coupled ZF-ND1 monomers, respectively. The total amount of transfected plasmid DNA in (**b**): L+R (V109K), 300 ng; L-P2A-R (V109K), 300 ng; L-T2A-R (V109K), 300 ng. The total amount of transfected plasmid DNA in (**c**): L+R (V109K), 300 ng; L+R (V109K), 150 ng; L-P2A-R (V109K), 150 ng; L-T2A-R (V109K), 150 ng.

**Figure 3 ijms-26-07602-f003:**
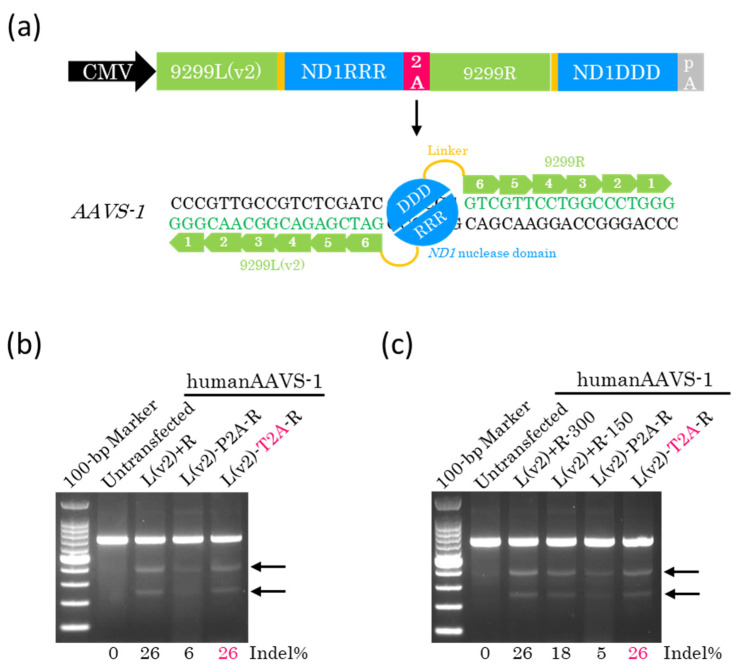
2A-coupled co-expression of two ZF-ND1 monomers on other ZF-ND1s. (**a**) The schematic of the 2A-coupled co-expression of two ZF-ND1 monomers targeting *AAVS-1*. 9299L(v2) and 9299R recognize their target DNA bases, and then the ND1 heterodimer nuclease (DDD/RRR) cleaves the spacer region between their recognition sites. (**b**,**c**) The T7EI assay. Gel images of T7EI-treated PCR products amplified from the target site. Arrows indicate the cleaved DNA band. L(v2)+R indicates the separately expressed ZF-ND1 monomers. L(v2)-P2A-R and L(v2)-T2A-R indicate the P2A- and T2A-coupled ZF-ND1 monomers, respectively. The total amount of transfected plasmid DNA in (**b**): L(v2)+R, 300 ng; L(v2)-P2A-R, 300 ng; L(v2)-T2A-R, 300 ng. The total amount of transfected plasmid DNA in (**c**): L(v2)+R, 300 ng; L(v2)+R, 150 ng; L(v2)-P2A-R, 150 ng; L(v2)-T2A-R, 150 ng.

## Data Availability

The data that support the findings of this study are available from the corresponding author upon reasonable request.

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
