# Peer review of "Efficient Genome Editing Using the T2A-Coupled Co-Expression of Two ZFN Monomers"

_ijms, 2025, doi:10.3390/ijms26157602_

Round 1

Reviewer 1 Report

Comments and Suggestions for Authors

The authors reported a design of ZFN effector expression using polycistronic cassette. The study was a step forward from their previous report on double plasmid co-expression of ZFNs, and showed promising results on the comparable or potentially better editing efficiency of the new designs.

Major comments:

The main place this paper can be improved on is the evidence supporting the “half-dose equal potency” claim.

Figure 2b 2c, and 3b 3c - the similar editing efficiency of 150ng and 300ng plasmid dose in the T2A/P2A groups indicates a saturation was reached for this assay, thus impact the ability for this assay to distinguish differences around double vs single plasmid. The authors should add a 150ng condition in the two-plasmid groups, to see if the editing potency drops more dramatically compares to the 2A design. I also suggest the authors to use a lower dose (i.e. 50ng vs 25ng+25ng) to perform this investigation. It’s possible the 2A designs will outperform the two-plasmid design at a lower dose.

Minor comments:

Line 30 - change “Cas9” to “SpCas9”

Line 42 - add reference #9 after “In a previous study”

Line 47 and through out the manuscript - remove “self-cleaving” because it’s reported that this is not the mechanism of action for the 2A peptides

Line 49-50 - please consider adding reference “Liu, Z., Chen, O., Wall, J.B.J. et al. Systematic comparison of 2A peptides for cloning multi-genes in a polycistronic vector. Sci Rep 7, 2193 (2017). https://doi.org/10.1038/s41598-017-02460-2”. It’s another comprehensive study on 2A peptides and their efficiency.

Reviewer 2 Report

Comments and Suggestions for Authors

I appreciate the work put into the manuscript titled "Efficient genome editing using the T2A-coupled co-expression 2 of two ZFN monomers." The authors demonstrated that NF-ND1 enables efficient gene editing in a 2A design, potentially streamlining its viral delivery. This study provides a valuable framework for future NF-ND1 development. However, its novelty is limited given that the 2A strategy was widely implemented in canonical ZFNs (ZF-FokI) years ago. Furthermore, the conclusion of T2A is superior for ZF-ND1 is only correct in HEK293T cells. It is still necessary to re-evaluate all 2A options for other cell types and further therapeutic applications.

Major Comments

  1. Abstract:
    The authors should highlight the use of ND1 nuclease in the abstract to avoid confusion with canonical ZFNs (ZF-FokI) and to emphasize the study's value.
  2. Introduction section lines 34-40:
    The authors should briefly mention the differences between ND1 nuclease and FokI nucleases. Additionally, they should acknowledge and cite the papers on the T2A strategy for ZF-FokI ZFNs in the introduction (not only in the discussion).
  3. Section title “T2A peptide is effective on other ZF-ND1”:
    This is misleading and could lead to an impression that T2A enables efficient gene editing on another engineered ZF-ND1. Please revise.
  4. Result lines 80-100 & lines 113-129:
    Almost two identical paragraphs with many repetitive sentences, for instance, “We transfected the plasmids into HEK293T cells, and 72 hours 115 after transfection, genomic DNA was extracted and subjected to a T7EI assay”. Please revise.
  5. Result & Discussion:
    The authors claim that the transfected DNA amount could be reduced by half in 2A designs. This may not be helpful if the cells are already saturated with plasmids. It would be beneficial to compare protein expression in different 2A conditions with two-plasmid systems. It's possible that 150 ng yields similar expression, suggesting plasmid saturation. Or that protein expression is lower in the T2A system, indicating that ZF-ND1 requires only minimal protein for genome editing.
  6. Discussion lines 141-145:
    The authors attribute the editing discrepancy to the 2A residue remaining on the protein. However, they lack sufficient information to draw this conclusion, as other factors, such as cell type-dependent variations in 2A efficiency, may influence editing efficiency. Therefore, stating that T2A is better suited than P2A for ZF-ND1 is incorrect.

Minor Comments

  1. Line 12: Please include the acronym of “transcription activator-like effector nucleases” (TALEN).
  2. Lines 38-40 & 58-60: Remove repetitive sentences.
Comments on the Quality of English Language

Many sentences need reorganization (e.g., lines 34-40).

Round 2

Reviewer 1 Report

Comments and Suggestions for Authors

The authors have addressed all my comments.

Reviewer 2 Report

Comments and Suggestions for Authors

Thank you for addressing my comments. The revised manuscript is clear, and I have no further concerns on this point.